Oceanographic habitat and the coral microbiomes of urban-impacted reefs

Rosales Stephanie M. 1 2
Sinigalliano Christopher 2
Gidley Maribeth 1 2
Jones Paul R. 1 2
http://orcid.org/0000-0003-4772-1991 Gramer Lewis J. 1 2 lew.gramer@noaa.gov
1 Cooperative Institute for Marine and Atmospheric Studies, University of Miami , Miami, FL , USA
2 Atlantic Oceanographic and Meteorological Laboratory, National Oceanic and Atmospheric Administration , Miami, FL , USA
Crandall Keith
Electronic publication date: 2019 Sep 10
Publication date: 2019
Volume: 7
Electronic Location ID: e7552
Received 2019 Apr 5; Accepted 2019 Jul 25
Copyright: © 2019 Rosales et al.
Copyright year: 2019
Copyright holder: Rosales et al.
License: This is an open access article distributed under the terms of the Creative Commons Attribution License, which permits unrestricted use, distribution, reproduction and adaptation in any medium and for any purpose provided that it is properly attributed. For attribution, the original author(s), title, publication source (PeerJ) and either DOI or URL of the article must be cited.
License URL: https://creativecommons.org/licenses/by/4.0/

Keywords: Currents, Bacteria, Waves, Temperature, Archaea, Turbidity, Tides, Oceanography

Funding: NOAA’s Coral Reef Conservation Program (CRCP Project #1114) NOAA ’Omics Initiative The study was funded by NOAA’s Coral Reef Conservation Program (CRCP Project #1114) and by the NOAA ’Omics Initiative. The funders had no role in study design, data collection and analysis, decision to publish, or preparation of the manuscript.

==============================
Coral reefs are in decline worldwide. In response to this habitat loss, there are efforts to grow, outplant, and restore corals in many regions. The physical oceanographic habitat of corals—such as sea temperature, waves, ocean currents, and available light—is spatially heterogeneous. We therefore hypothesize that outplant location may affect microbiomes, and ultimately, coral health and restoration success. We evaluated the influence of the physical oceanographic habitat on microbes in wild Porites astreoides and Siderastrea siderea. Tissue samples were collected at four Florida reefs in March, June, and September of 2015. We estimated oceanographic conditions from moored instruments, diver observations, remote sensing data, and numerical models. We analyzed microbiomes using amplicon 16S rRNA high-throughput sequencing data. We found microbial alpha-diversity negatively correlated with in situ sea temperature (which represented both the annual cycle and upwelling), as well as modeled alongshore currents, in situ sea-level, and modeled tide. Microbial beta-diversity correlated positively with significant wave height and alongshore currents from models, remotely-sensed relative turbidity, and in situ temperature. We found that archaea from the order Marine Group II decrease with increases in significant wave height, suggesting that this taxon may be influenced by waves. Also, during times of high wave activity, the relative abundance of bacteria from the order Flavobacteriales increases, which may be due to resuspension and cross-shelf transport of sediments. We also found that bacteria from the order SAR86 increase in relative abundance with increased temperature, which suggests that this taxon may play a role in the coral microbiome during periods of higher temperature. Overall, we find that physical oceanographic variability correlates with the structure of these coral microbiomes in ways that could be significant to coral health.

Background

Coral reefs face numerous environmental and biological challenges, especially in a changing climate. Understanding how the physical habitat—such as temperature variability, light, and water flow—interacts with biological factors in corals has important implications for managing these sensitive ecosystems. Environmental factors such as wind and surface ocean waves can contribute to the spatial morphology of coral reefs over time (Gischler, Storz & Schmitt, 2014). Waves help form organismal and morphometric gradients on reefs (Harborne et al., 2006; Kelly et al., 2014; Kench & Brander, 2006; Porter, Branch & Sink, 2017), driving both coral breakage and intermittent water flow on shallow reefs. Coral health is dependent as well, on geographically fine-scale ocean circulation, which varies inter-annually, seasonally, and over shorter time scales (Davis et al., 2011; Feng et al., 2016a; Monismith et al., 2018; Rogers et al., 2013; Shedrawi et al., 2017; Sponaugle et al., 2005). For example, coral larval dispersion and settlement rates are affected by hydrodynamic variability, which ultimately impacts gene flow and the genetic diversity of coral reefs (Baums, Paris & Chérubin, 2006; Serrano et al., 2014; Serrano et al., 2016).

Oceanographic processes can also affect the distribution of nutrients across a reef (Zhang et al., 2011). Currents and upwelling can bring nutrients from the ocean to reefs, and also flush water that has been depleted of nutrients. Nutrient uptake by corals is also augmented by increased wave orbital motion and wave-driven circulation (Hearn, Atkinson & Falter, 2001; Lowe & Falter, 2015), especially in shallow reefs (Zhang et al., 2011). Additionally, upwelling on coral reefs—the upward, onshore motion of cooler waters from the subsurface ocean (Gramer et al., 2018; Leichter et al., 1996)—can aid in both survival and recovery from a bleaching event (Bayraktarov et al., 2013; Tkachenko & Soong, 2017). For instance, in Tayrona National Natural Park in Colombia, corals that underwent bleaching recovered by 97% in areas where seasonal upwelling occurred (Bayraktarov et al., 2013).

The health of corals is associated with the structure and composition of their microbial communities. With healthy corals showing microbial patterns distinct from diseased or stressed colonies (Glasl, Herndl & Frade, 2016; Kimes et al., 2010; Roder et al., 2014). Coral microbiomes respond to both biotic and abiotic fluctuations. While many studies have examined the effects of temperature on coral microbiomes (Boehm et al., 2004; Feng et al., 2016b; Bourne et al., 2008; McDevitt-Irwin et al., 2017; Zaneveld et al., 2016; Ziegler et al., 2017), fewer studies have examined the relationship of coral microbiomes to other physical oceanographic variables, such as water flow (including tides and wave-driven flow) and available light (or turbidity). A recent study evaluated the changes in microbial communities in the coral Coelastrea aspera during a 4 days spring tide event in Phuket, Thailand (Sweet et al., 2017). It was observed that spring tides induced changes in microbial composition that were likely driven by transient microbial members. Another recent study showed that Acropora muricata corals exposed to higher rates of water flow in both tank and in situ experiments have a more stable microbiome (Lee et al., 2017). Coral reefs with higher impacts from turbidity showed a relative increase in opportunistic bacterial pathogens, and a decrease in symbiotic bacteria (Ziegler et al., 2016). Also, one recent study found that temperature and available light affected prokaryotic communities associated with the tissue of Pocillopora damicornis, a common brooding coral in the Indo-Pacific (Van Oppen et al., 2018).

Studies like those highlighted above, provide evidence that the physical habitat can impact coral microbial community members and suggests that this may affect the health of the host. In 2008, it was estimated that southeast Florida reefs had less than 5% live coral cover, compared to higher values in prior decades (Gilliam, 2009). As a result of this ongoing decline (Lirman et al., 2011), there are now efforts to restore and outplant corals in Florida (Lirman & Schopmeyer, 2016; Young, Schopmeyer & Lirman, 2012). Both site selection and host genetics have been shown to be strong indicators for growth in outplanted corals (Drury et al., 2017). Nonetheless, there is still a lack of information in the literature on precisely which physical habitat characteristics can increase the incidence of beneficial coral microbes and ultimately increase rates of coral survival (Glasl, Herndl & Frade, 2016).

In our study, we aimed to identify the impacts of the physical oceanographic habitat—in particular, turbidity, temperature, waterflow (currents, waves, tides), and bathymetry—on coral microbiomes. We used a bioinformatics approach to evaluate microbial communities using 16S rRNA amplicon high-throughput sequencing on samples collected in March, June, and September of 2015. These samples were collected from the tissue of two stony coral species in the wild, Porites astreoides—a brooding coral considered resilient, but currently being affected by disease in Florida, and Siderastrea siderea—a reef-building broadcast spawner also being affected by disease (Edmunds, 2010; Walton, Hayes & Gilliam, 2018).

Corals were collected from four southeast Florida reefs: Barracuda, Oakland Ridge, Emerald, and Pillars (Table 1; Fig. 1). Physical habitat variables were investigated after sample collections were completed. We used longitude and latitude coordinates and dates that coral tissue was sampled to extract oceanographic habitat data from an operational wave model, a quasi-operational ocean-circulation model, and satellite remote sensing products. In-water measurements of temperature were recorded at the sampling sites. Seafloor moorings near the study region collected data on current profiles, sea temperature, and seafloor pressure.

Figure 1 Geography of the study region.

(A) Wider geography of the Florida reef tract, showing boundaries of southern Florida counties. Sampling sites are shown as colored markers offshore of Miami-Dade and Broward counties. Gray contours show the 10, 30, and 250 m isobath. (B) High resolution bathymetry of the sampling region, showing southeast Florida’s coastline in heavy black, sampling sites as colored markers, and isobaths as color contours at three m depth intervals down to 30 m, followed by the 40, 50, and 60 m isobaths offshore. The high resolution of this bathymetry highlights the linear, north-south features of south Florida reefs, as well as major man-made channels cut across them: Port of Miami at 25.76, Haulover at 25.90, and Port Everglades at 26.09°N latitude.

Table 1 List of categorical variables and continuous variable medians that were correlated with coral microbiota.

Reef	Month	Species	N	% coral cover cm2	µ coral size cm3	% coral cover for SS and PA cm2	µ coral size for SS and PA cm3	Depth (m)	Sea floor slope	Min. dive watch temp (°C)	Distance to nearest inlet (km)	Relative turbidity	Significant wave height (m)	Mm tide stage (m)	Sea level (m)	Along-shore current (m/s)	Cross-shore current (m/s)	
Barracuda	March	PA	2	1.97	354	0.209	130.1	−9.76	0.037	23.2	2.470	−0.068	0.277	−0.0080	−0.050	0.346	−0.050	
Barracuda	June	PA	2	2.01	343	0.264	129.75	−9.76	0.037	28.2	2.470	0.142	0.180	0.0008	0.007	0.475	−0.077	
Barracuda	September	PA	2	1.5	333.8	0.262	132.73	−9.76	0.037	29.2	2.470	−0.478	0.074	−0.0058	−0.091	0.329	−0.084	
Barracuda	March	SS	4	1.97	354	0.209	156.33	−10.07	0.031	23.4	2.029	−0.060	0.254	−0.0080	−0.050	0.341	−0.045	
Barracuda	June	SS	2	2.01	343	0.264	129.75	−10.15	0.031	28.2	2.317	0.143	0.176	0.0008	0.007	0.470	−0.081	
Barracuda	September	SS	4	1.5	333.8	0.262	127.25	−10.07	0.031	29	2.029	−0.383	0.068	−0.0058	−0.091	0.325	−0.072	
Emerald	March	PA	1	1.33	261.2	0.169	232.33	−7.25	0.024	24	10.183	−0.075	0.258	−0.0052	−0.075	0.445	0.036	
Emerald	June	PA	1	1.28	209.5	0.193	93.75	−7.25	0.024	28	10.183	−0.426	0.182	0.0016	0.064	0.614	0.040	
Emerald	September	PA	3	1.17	198.8	0.205	230.5	−8.48	0.022	29.2	10.528	−0.379	0.225	0.0003	0.155	0.377	0.031	
Emerald	March	SS	1	1.33	261.2	0.169	101.86	−7.25	0.024	24	10.183	−0.075	0.258	−0.0052	−0.075	0.445	0.036	
Emerald	June	SS	1	1.28	209.5	0.193	93.75	−7.25	0.024	28	10.183	−0.426	0.182	0.0016	0.064	0.614	0.040	
Emerald	September	SS	3	1.17	198.8	0.205	87.5	−8.48	0.022	29.2	10.528	−0.379	0.225	0.0003	0.155	0.377	0.031	
Oakland Ridge	March	PA	3	2.34	186.2	0.463	223.5	−8.59	0.048	22.4	7.206	−0.089	0.255	−0.0076	−0.049	0.259	−0.001	
Oakland Ridge	June	PA	1	2.37	199.9	0.519	161.71	−8.67	0.048	28	7.206	0.110	0.143	−0.0050	−0.104	0.302	0.026	
Oakland Ridge	September	PA	3	2.31	193.8	0.519	219.67	−8.59	0.048	28.8	7.206	−0.141	0.067	−0.0043	−0.088	0.245	0.023	
Oakland Ridge	March	SS	3	2.34	186.2	0.463	171.07	−8.59	0.048	22.4	7.206	−0.089	0.255	−0.0076	−0.049	0.259	−0.001	
Oakland Ridge	September	SS	3	2.31	193.8	0.519	161.71	−8.59	0.048	28.8	7.206	−0.141	0.067	−0.0043	−0.088	0.245	0.023	
Pillars	March	PA	2	1.36	244.5	0.192	126.5	−10.31	0.014	23.2	4.784	−0.128	0.310	−0.0065	−0.062	0.310	−0.024	
Pillars	June	PA	2	1.03	300.9	0.134	162	−10.31	0.014	28	4.784	−0.164	0.183	0.0021	0.075	0.406	−0.021	
Pillars	September	PA	2	0.97	290.3	0.149	101	−10.31	0.014	30.2	4.784	−0.089	0.253	0.0005	0.155	0.238	0.000	
Pillars	March	SS	4	1.36	244.5	0.192	269	−10.92	0.015	23.8	4.531	−0.121	0.327	−0.0065	−0.062	0.310	−0.006	
Pillars	June	SS	2	1.03	300.9	0.134	162	−11.49	0.015	28	4.412	−0.172	0.201	0.0021	0.075	0.405	0.007	
Pillars	September	SS	4	0.97	290.3	0.149	205.6	−10.92	0.015	30	4.531	−0.088	0.267	0.0005	0.155	0.238	0.013	
Notes:

Only % coral cover and mean coral size of the overall reef were not used in the analysis.

N is the number of replicates (distinct coral heads) of that host species sampled at that reef in that month. SS = Siderastrea siderea and PA = Porites astreoides.

In this paper, we summarize variability in the physical habitat across the three sampling months and the four reefs. We then seek correlations between the physical habitat and coral microbiome data using stepwise linear regression models. Our data show that independent of the physical habitat, coral-host species had a strong correlation with microbial composition. Yet, for both host species, physical oceanographic variables, as well as month of sampling and reef location, were all significant in structuring microbiomes. Our analysis suggests that the physical habitat may contribute to changes in the coral microbial community over both time and space.

Methods

Study region

The Florida reef tract (FRT) is a coral reef system in the USA, that stretches from the Dry Tortugas (83.0W, 24.5N) east and north to the lower edge of the South Atlantic Bight (80.0W, 27.3N). The northern third of the FRT lies along the southeast Florida shelf, offshore of Miami-Dade, Broward, and two other counties. For this study, four reefs were sampled (Fig. 1): Barracuda and Oakland Ridge reefs in Broward, and Emerald and Pillars reefs in Miami-Dade. Three distinct sites within each reef were selected for replicate coral tissue collections, resulting in a total of 12 sampling sites. These sites are in waters between 6 and 12 m deep, and lie between 1.1 and 5.5 km offshore of mainland Florida.

Sample collection

The data for this study were leveraged from previous projects. Details about the sites and collection methods have previously been presented (Sinigalliano et al., 2018; Staley et al., 2017). Briefly, samples for this study were collected during March, June, and September of 2015 (see Table 1). Visually healthy corals were sampled at random from two coral species, Porites astreoides and S. siderea, with SCUBA and a syringe biopsy method. This entailed loosening the polyps from the calyx by gently scraping/digging at the polyp tissue before adhering a five mL syringe against the coral polyp to create a suction and extract that polyp from the coral head. This was repeated twice per coral head and combined into one sample in the lab. When divers reached the boat, samples were transferred to a 50 mL tube with 45 mL of 95% molecular grade ethanol, placed on ice, then preserved at −80 °C at the NOAA AOML laboratory. In addition, divers measured sea temperature with a dive-watch (see Physical oceanographic habitat analysis), and both overall and per-species coral cover and coral mean size (Sinigalliano et al., 2018) at each of the reef sites. A total of 24 Porites astreoides and 31 S. siderea samples were collected across four reefs (Barracuda, Oakland Ridge, Emerald, and Pillars) during the 3 months of sampling (March, June, and September).

DNA extractions and microbiome amplicon sequencing

Coral polyps, which included the tissue and remnant mucus, were processed for DNA extraction and sequencing, briefly as follows (Staley et al., 2017): samples were vortexed to resuspend coral tissue, and the entire contents were placed on a 0.2 µm polycarbonate filter under a vacuum. Both the filter and any residual large coral tissue were transferred to a bead beating Lysing Matrix “E” tube. The samples were then homogenized by bead beating with a FastPrep 24 instrument (MP, Biomedicals, Irvine, CA, USA) by two separate rounds of bead beating at an impact speed of 6.0 m/s for 60 s each (for a total of 120 s at 6 m/s). Each lysed sample was then purified with the FastDNA SPIN kit for soil DNA Extraction (MP, Biomedical) following the manufacturer’s instructions and eluted in 100 μL of TE buffer. DNA was amplified using dual index primers, 515F and 806R, that target the variable-4 (V4) region of the 16S rRNA gene. Amplified DNA was sequenced on the MiSeq platform, generating paired-end reads of 300 base pairs (bps).

Bioinformatic analysis

Demultiplexed sequences from the 55 samples were retrieved from the National Center for Biotechnology Information (NCBI) Sequence Read Archive database (#SRP076111). Once downloaded, the sequence primers were trimmed using the program cutadapt. The sequences were analyzed with qiime2-2018.6 using the DADA2 pipeline for classification of amplicon sequence variants (ASVs), and were run with a max expected error of 2 (Callahan et al., 2016). To select sequence trimming positions, the quality score of a random set of sequences were analyzed. Where sequences showed a drop-in quality score (<Phred = 30), those bp positions where chosen for trimming. All forward sequences were truncated to the same length of 220 bps and trimmed at the first eight bps. The reserve reads were also truncated to be of the same length at 120 bps, and trimmed at the first 20 bps. The DADA2 pipeline was used for ASV selection, dereplication, merging paired-end reads (min overlap = 20 bps) and removing chimeras (with the “consensus” option). Taxonomy was assigned with the Silva database version132 training set, which was trained on the V4 region by the fit-classifier classify-sklearn function. Only sequences that were taxonomically identified as Bacteria or Archaea were analyzed. The biome data, ASV table, fasta file, phylogenetic tree, and metadata for this analysis can be found on figshare (DOI 10.6084/m9.figshare.7388672). The specific NCBI accession number for each sequence file is listed under “sampleID” in the metadata file.

Statistical analysis

Microbial alpha-diversity was evaluated with qiime2-2018.6, using the Shannon diversity index and species evenness metrics. To assess alpha-diversity, ASV counts were rarefied to (randomly filtered to an equal sequencing depth of) 403 counts, so that all 55 samples could be evaluated. Since 403 is a low number of sequences, the ASV table was also rarefied to 2,055 (N = 53). We compared these two methods and since both methods resulted in the same conclusions, a rarefaction of 403 was used in order to include all 55 samples in alpha-diversity analysis. Categorical habitat data (see Table 1) were tested for significant relationships with Shannon diversity index and species evenness using the alpha-group-significance function, which applies a pairwise/all-group Kruskal–Wallis test (Gregr & Bodtker, 2007). Continuous habitat data (see Table 1) were tested vs. alpha-diversity with a simple linear regression, using the function lm on R.3.4.3 (R Core Team, 2017).

For beta-diversity tests, the data were not rarefied, but instead were filtered by retaining only those taxa that were seen four or more times in at least 10% of the samples. The counts were then transformed into relative abundances. Continuous habitat data were evaluated for pairwise covariance; from each pair with a correlation >80%, one variable was selected for further analysis. Principal coordinate analysis (PCoA) ordination was applied to the relative abundance of ASVs with a weighted UniFrac distance metric (Lozupone & Knight, 2005). Discrete variables (i.e., reef site, species, and month) were tested using Analysis of similarities (ANOSIM) and a Permutational multivariate analysis of variance (PERMANOVA) with 999 permutations to identify which groups were different (Clarke & Warwick, 1994). The counts matrix was also used to build a model with forward and backward model selection, using the vegan function ordistep, which selects significant variables that best explain the counts data. Prior to inputting habitat variables into ordistep, their respective variance-inflation factors (Salmerón, García & García, 2018) (VIF) were tested; variables with a VIF score >20 were removed from further analysis. The variables selected by the model were then used to construct a canonical or constrained correspondence analysis (CCA) (Henriques et al., 2006). These statistical analyses were conducted with the packages vegan 2.4.6 and phyloseq 1.22.3.

To further evaluate patterns seen in the beta-diversity results, we used a classification method to evaluate which microbial taxa were important to the physical habitat variables that were found to be significant using ordistep (above). The classification method used on each of the significant physical habitat variables was the sample-classifier regress-samples function from qiime2. This function performs random forest (a supervised machine learning algorithm) and was used with parameters --p-parameter-tuning, --p-estimator RandomForestRegressor (RFR), and --p-n-estimators 500. The model outputs importance values and the highest importance values have the greatest prediction power. Thus, the five taxa with the highest importance values from each random forest test were selected for simple linear regression analysis vs. the corresponding physical habitat variable. Only the taxa with a significant correlation (p < 0.05) and a coefficient of determination R2 > 0.2 were plotted. The code for the statistical analysis for the microbial data and the corresponding figures are found on figshare (DOI 10.6084/m9.figshare.7925495).

Physical oceanographic habitat analysis

The minimum near-bottom sea temperatures were collected by divers (see Sample collection). Replicates for each measurement are listed in Table 1. Seafloor depths (bathymetry) were derived from NOAA’s 10 m horizontal-resolution port-bathymetry project (Carignan et al., 2015). From bathymetry, we estimated local seafloor depth ([m]), seafloor slope (rise/run), isobath orientation (alongshore direction, deg. True), and distance from shore. We also estimated distance from each sampling site to the nearest major inlet running through the Intracoastal Waterway (Fig. 1; Table 1).

Satellite multispectral ocean-color was used to produce a proxy for in-water turbidity, a product called relative turbidity or color index (CI). Since 2000, the MODerate-resolution Imaging Spectrophotometer instruments aboard the polar-orbiting Aqua and Terra satellites have produced multispectral ocean color data at 250 m horizontal resolution. CI datasets were produced from these data, using an algorithm developed by the University of South Florida’s Optical Oceanography Laboratory (Barnes et al., 2015).

The CI product is subject to bias in shallow waters where the seafloor may be visible from space. In this study, CI values were normalized using the median and 7th and 93rd percentiles for 15 years of CI data (2000–2015) from each pixel. Percentile values were used because CI for coastal pixels is often not normally distributed (Gramer & Hendee, 2018). On days when clouds obscured a pixel during one or both daytime satellite overpasses, no CI value was available for that site. Therefore, for the purposes of this study, we used a 14 days arithmetic mean of CI values centered on each sample date, resulting in relative turbidity time series with no gaps (Figs. 2A–2C). Finally, time series for satellite CI were validated by linear regression vs. in-water turbidity measurements (NTU) done with boat-based casts using a factory-calibrated nephelometer throughout the sampling period (2013–2015) (Fig. S1; Staley et al., 2017).

Figure 2 Physical habitat spatial fields, with time-averaged across the full weeks surrounding each sampling period.

In (A and D) March, (B and E) June, and (C and F) September. (A–C) show normalized relative turbidity from satellite, and (D–F) show alongshore current (m s−1) from GoM HYCOM. In (D–F), alongshore current speed (v) is shown as colored contours, and overall current direction is shown as scaled black arrows at the native model resolution. Locations of the 12 individual coral sampling sites are shown with markers, colored by reef name.

Ocean currents were extracted as daily snapshots from the Gulf of Mexico HYbrid-Coordinate Ocean Model (GoM HYCOM; Gierach, Subrahmanyam & Thoppil, 2009) at the vertical layer closest to the surface. Northward and eastward vector-components were bilinearly interpolated from native model resolution (1/25th degree) to individual sampling location. Current vectors were then rotated so that positive cross-shore currents were directed perpendicular to the local isobath (i.e., “offshore”) at each site. Daily snapshots of currents from GoM HYCOM alias the diurnal and semidiurnal tidal cycles, showing the distinct signature of the 27.55 days Mm tidal constituent. Furthermore, although we expect GoM HYCOM to provide useful statistics for the amplitude of lower-frequency (multi-day) current variability, the model may not always correctly model the phase of that variability (Gramer, 2013). Thus, centered moving 14 days averages of each current component (cross-shore and alongshore) were analyzed (Figs. 2D–2F).

To estimate surface wave action, we used significant wave height from NOAA’s WaveWatch III multi-mesh grid operational model for the western Atlantic, with four nautical-mile nominal resolution (Lee et al., 2009; Tolman, 2014). A wave attenuation algorithm was applied to wave height, bilinearly interpolated to the horizontal resolution of the bathymetry (10 m) (Fig. S2; Gramer & Hendee, 2018). This algorithm linearly reduces significant wave height at bottom-depths between 0 and 20 m, resulting in high-resolution wave fields that reach all the way inshore, with zero wave-height at the beach (Fig. S3).

Ocean model outputs and diver-watch sea temperatures were validated using in situ hourly observations from acoustic Doppler current profilers (ADCP) (Fig. S4). These ADCPs were moored at two sites, one at 7 and one at 26 m depth, six km north of Barracuda Reef (Sinigalliano et al., 2018). They measured near-bottom sea temperatures (nominal accuracy 0.01 K) and three-dimensional ocean current profiles (bin sizes 0.5 m at 7 m depth and 1 m at 26 m, nominal accuracy 10−4 m s−1) every 20 min throughout all three sampling periods of March, June, and September of 2015.

Tide heights for each site were modeled using the TPXO 7.2 global tide solution (Stammer et al., 2014). Hourly in situ measurements of near-bottom sea-pressure from the FACE mooring at seven m depth were also used with mean depth removed (“sea-level anomaly”), both to validate model tide-stage predictions and to consider the additional effects of setup from wind, waves, and larger-scale sea-level variation offshore. The time of day of collection was not logged for the tissue samples analyzed in this study, nor can the TPXO 7.2 global solution model fine-scale variability in tide heights and currents on the topographically complex southeast Florida shelf (~1 km, diurnal or faster). For these reasons, a 14 days centered average of both the modeled tide height and the sea-level anomaly (Fig. S5) were analyzed in the present study. The Ecoforecasts toolkit, a set of MATLAB functions developed by Gramer as a part of NOAA CHAMP, was used to produce some time series analyses and maps presented here. Due to the prevalence of non-Gaussian distributions in habitat data (Fig. S6), results for habitat variables are cited as median ± half of interquartile range values. Elsewhere, for example, for relative abundances, mean and standard deviation are reported. The code used to generate physical ocean habitat variables and the corresponding figures are found on figshare (DOI 10.6084/m9.figshare.7925546).

Results

The physical oceanographic habitat of four southeast Florida coral reefs in 2015

To extract all other physical habitat variables for each of the four reefs, we used longitude and latitude coordinates from each of the 12 sample-collection sites (three sites per reef, Fig. 1) together with the dates of sampling, as described in the Methods section. March was a period of geographic patchiness in CI (Fig. 2A). However, patchiness was largely confined during March sampling dates to waters away from the sampling sites. On the other hand, both June and September (Figs. 2B and 2C) showed less intense regional patchiness, but patches of higher relative turbidity did reach some of the sites during sampling. Satellite relative turbidity (CI) showed a complex seasonal pattern than the other variables considered (Fig. 3C). The highest extreme and the greatest range of values occurred during June sampling (−0.43–0.23, median −0.16), with sampling in September scattered over a mainly lower range (−0.50 to −0.08, median −0.23), and that from March clustered within a very narrow range (−0.13 to −0.04, median −0.09). Barracuda was the reef that experienced the greatest positive and negative extremes of normalized CI during the 2015 sampling, with a peak at the three Barracuda sites of between +0.06 and +0.23 on the sampling day of June 12th, and lows between −0.30 and −0.50 on September 11th. Interestingly, June was also the month with the highest CI value at Oakland Ridge (0.11), despite lower modeled waves. (Results of regressing satellite CI values vs. in-water turbidity measurements are shown in Fig. S1)

Figure 3 Physical habitat time series for 2015 at four reefs.

Reefs are marked by different shapes and dates of sampling are shown as colored markers: (A) per-dive minimum measured sea temperature (°C) from divers, and for comparison, from in situ moorings inshore of (black line, at seven m depth) and offshore of (gray line, at 26 m depth) the reef lines, as well as from a dockside station (dark red line, depth about two m) approximately 10 km inshore. (B) The 14 days centered moving average of significant wave height (m) from NOAA WaveWatch III surface wave model, at the grid point where each sample was gathered. (C) The 14 days centered moving average of normalized relative turbidity from satellite ocean color at the pixel where each sample was gathered. (D) The 14 days centered moving average of alongshore current (m s−1) from GoM HYCOM linearly interpolated to the location where each sample was gathered. The Low-pass (40-h) filtered time series (40 HLP) of measured depth-averaged ocean currents are shown in black for moorings immediately inshore (seven m depth, solid) and offshore (26 m depth, dashed) of the major reef line.

Alongshore currents from the Gulf of Mexico HYCOM model (Figs. 2D–2F and 3D) showed a relative peak at all sampling sites during June, with weekly medians of +0.30 to +0.65 m s−1 (positive being a northward current). Further peaks in July and August were followed by a rapid decline during sampling weeks in September, to medians of +0.10 to +0.45, which was also similar to medians during the March sampling. At the time of coral tissue sampling, in situ ocean current profilers were deployed near the sampling sites. The midsummer peak could also be seen in these instruments (Fig. 3D, gray and black lines). According to climatology models (Fig. 2E), this summer peak corresponded to a westward meander of the Florida Current; this June meander however, affected the southernmost sampling sites, those at Emerald Reef (median +0.60 ± 0.00 m s−1), more than at the other three reefs (medians +0.30 to +0.48 ± 0.01 m s−1). Emerald Reef actually experienced higher alongshore currents than the other sites during all three sampling months, although these differences were less extreme in March and September than they were in June.

The minimum diver-watch sea temperature was recorded during each tissue-sampling dive (Table 1). For validation, dive-watch temperatures were compared to sea temperatures measured at two moorings (that at 7 and that at 26 m depth). Dive-watch sea temperature followed closely the coincident mooring temperatures at seven m (Fig. 3A; black line): regressions were significant (p < 0.0001, Fig. S4), with root mean-squared error (RMSE) 0.28 at Barracuda, 0.78 at Oakland Ridge, 0.31 at Pillars, and 0.29 K at Emerald. Regressions vs. the deeper mooring at 26 m were also significant (p < 0.05) with RMSE of 0.8 at Barracuda, 1.0 at Oakland, 0.34 at Pillars, and 0.25 K at Emerald.

Like the moorings, dive-watch sea temperature across the four reefs followed a seasonal pattern (Fig. 3A). For March, there was a median 23.2 and interquartile range ± 0.4, for June 28.0 ± 0.2, and for September 29.2 ± 0.5 °C. However, temperatures were somewhat cooler than might be expected from historical in situ measurements in this region, especially during the warmer months (Gramer et al., 2018). In particular, the mooring at seven m depth (Fig. 3A) recorded near-bottom temperatures of 24.7 ± 0.5 in March, 29.2 ± 0.9 in June, and 31.2 ± 0.4 °C in September of 2015. Similarly, near-surface temperature measurements in two m of water at the nearby Virginia Key pier (NOAA National Ocean Service station “VAKF1”) showed warmer temperature ranges in June and September of 29.7 ± 1.1 and 30.1 ± 0.6 °C, respectively (Fig. 3A; red line).

Wave heights from the attenuated NOAA model showed a relatively simple seasonal pattern for all sites (Fig. 3B), with the highest waves at all sites during sampling in March (0.25–0.35 m), and intermediate waves during sampling in late September (0.2–0.3 m). Waves in June were generally lower (0.15–0.25 m), due to the absence of storm systems in the nearby Atlantic and Gulf of Mexico, and to generally light local summer winds. The lowest wave estimates, however, were those for samples taken in early September at Oakland Ridge and Barracuda (0.08 m); these two reefs were generally the most sheltered (Fig. 1; Fig. S3). Furthermore, these samples were gathered prior to a series of wind events in mid-September, and perhaps more importantly, prior to the development of a major tropical weather system in the Atlantic in late September (Fig. 3B; see the Discussion section).

Tide models used for this study do not provide sufficient spatial or phase resolution to estimate diurnal or semidiurnal variability in water heights or tidal currents. However, these models do resolve variability between sites associated with the Mm tidal constituent, which has a 27.55 days period. Tide height variations for this constituent were at or near their minimum (−0.008 to −0.005 m) during all March sampling dates. In June, sampling was done near the Mm peak (+0.002 m) at all reefs except Oakland Ridge, where samples were gathered near the time of the minimum or ebb Mm (−0.005 m). Similarly, September samples were collected near the ebb at both Oakland Ridge and Barracuda (−0.006 to −0.004 m), and near the peak at Pillars and Emerald Reef (+0.001 m).

Unlike with the tide model, sea-level anomaly as estimated from the seven m mooring shows the effect of sea-level setup from wind and possibly other forcings, in addition to the effect of the barotropic tide (Cushman-Roisin & Beckers, 2011). A sea-level anomaly in March at all reefs clustered between −0.04 and −0.08, and in September, it clustered even more tightly at −0.09 for most reefs, with Pillars the sole September outlier at +0.16 m. Finally, in June, sea-level anomaly ranged more widely, from a high of +0.08 at Pillars, to +0.06 at Emerald, +0.01 at Barracuda, and −0.10 m at Oakland Ridge.

Fictibacillus and Endozoicomonas were dominant bacterial members in two hard coral species in four southeast Florida reefs

The 55 16S rRNA amplicon sequences from coral tissue samples (24 Porites astreoides and 31 S. siderea) were leveraged from a previous study (Staley et al., 2017). The program DADA2 identified a total of 22,451 ASVs and after filtering, 15,201 ASVs remained. The minimum frequency of an ASV per sample was 404 and the maximum frequency was 196,536. The median ASV frequency per sample was 102,667 with a mean of 98,118.

The microbial taxa of the samples are summarized in Fig. 4. On average, across the four Southeast Florida reefs in 2015, the bacterial genus Fictibacillus, family Bacillaceae made up the majority of the microbial community (N = 55; mean relative abundance, meanRA = 13.9%, ±standard deviation 1.8%). The bacterial genus Endozoicomonas, family Endozoicomonadaceae also had a high meanRA (12.2% ± 9.7%). When analyzing the two host-species separately, Porites astreoides was dominated by Endozoicomonas (N = 24; 10.7% ± 8.0%); S. siderea on the other hand, had a higher abundance of the genus Fictibacillus (N = 31; 10.4% ± 11.1%), while Endozoicomonas was only this host’s 9th most abundant taxon (1.5% ± 3.0%). During the month of September (N = 24) specifically, across samples from both host species, the family Bacillaceae was found at higher abundances, and was composed of the genera Paenibacillus (8.9% ± 12.2%) and Fictibacillus (7.7% ± 9.6%). However, for the months of March (N = 20) and June (N = 11), across all sites and both host species, Endozoicomonas showed the highest meanRA of 4.5% ± 5.2% (March) and 5.2% ± 9.0% (June). Two reef sites had the highest overall average abundances for Fictibacillus across all months and both hosts: Barracuda (N = 16; 4.3% ± 10.0%) and Emerald (N = 10; 3.8% ± 9.6%). The other two reefs showed their highest overall average abundances for Endozoicomonas across month and host: Oakland Ridge (N = 13; 4.9% ± 8.0%) and Pillars (N = 16; 4.3% ± 6.8%).

Figure 4 Relative abundance of microbial communities of Siderastrea siderea and Porites astreoides tissue from four Southeast Florida reefs.

Stacked bar colors indicate different genera and the relative abundance of each genus of the most abundant (>0.05%) ASVs. The figure is grouped by reef site, the month of sampling, and the host-species.

Bacterial alpha diversity correlated with the physical oceanographic habitat

Table 1 lists three (3) categorical and 14 numerical variables. A total of 12 physical habitat variables were selected for analysis to identify univariate correlations between these variables and coral microbiomes (Overall coral cover and overall mean size were not considered; only coral cover and mean size for the two host species were considered - Porites astreoides and S. siderea.). Latitude and longitude were removed from this analysis, as they correlated (ρ > 80%) with other independent habitat variables.

Microbial Shannon diversity was negatively correlated with alongshore current speed (R2 = 0.05, p = 0.044), tide height (R2 = 0.12, p = 0.004), and temperature (R2 = 0.13, p = 0.002; Figs. 5A–5C). Similarly, microbial evenness was negatively correlated with sea-level (R2 = 0.09, p = 0.014), tide height (R2 = 0.12, p = 0.005), and temperature (R2 = 0.15, p = 0.002; Figs. 5D–5F). Among independent categorical variables, host species was significant to Shannon diversity (p = 0.04). With regard to interactions between sampling months, pairwise comparisons were significant for both Shannon and evenness (p < 0.05) for March vs. June and March vs. September, but not for June vs. September. Finally, there were no significant pairwise differences between reefs, in either Shannon diversity or evenness.

Figure 5 Microbial alpha-diversity from coral species Siderastrea siderea and Porites astreoides from four Southeast Florida reefs.

Samples are colored by month and shaped by the reef name. The blue line represents the linear regression best fit between the habitat variable (x-axis) and the diversity metric (y-axis); gray shadow around the blue line is the 95% confidence interval. The first column plots Shannon diversity vs. physical habitat variables that were found to be significant to it (p < 0.05) (A) alongshore currents (m s−1), (B) tide (m), and (C) temperature (°C). The second column of plots show species evenness of habitat vs. variables that were significant (p < 0.05) (D) sea-level anomaly (m), (E) tide (m), and (F) temperature (°C).

Coral bacterial community composition correlated with physical oceanographic habitat

In the beta-diversity analysis, ordination axis 1 of the PCoA explained 89.5% of the variance, and PCoA 2 explained 5.8% of the variance (Fig. 6A). Of correlations between the 12 numeric habitat variables considered, and axis 1 or axis 2 of the ordination, only relative turbidity was significant to axis 2 (R2 = 0.27, p < 0.05). In addition, for categorical variables, an ANOSIM showed that the microbial community correlated with host species (R2 = 0.38, p = 0.001) and month (R2 = 0.21, p = 0.001), but not with reef location (R2 = 0.04, p = 0.094). A PERMANOVA also showed the same results for host-species (p > 0.001), month (p > 0.001), and reef location (p = 0.86). Four of the 12 habitat variables had explanatory power for the microbiome count matrix: alongshore currents (p = 0.005), relative turbidity (p = 0.015), significant wave height (p = 0.005), and temperature (p = 0.005). All four of these variables had a VIF (diagonal of the inverse of the correlation coefficient matrix) <2, indicating that they represent effects which are independent of one another (Salmerón, García & García, 2018).

Figure 6 Microbial beta-diversity of coral Siderastrea siderea (SS) and Porites astreoides (PA) tissue from four Southeast Florida reefs.

(A) Principal coordinate analysis (PCoA) with a weighted UniFrac distance, colored by coral host-species, labeled by the month of collections and shapes are based on reef location. (B) Constrained correspondence analysis (CCA) with a Bray–Curtis statistic for microbial compositional dissimilarity. The microbiome correspondence analysis was constrained by temperature (°C), alongshore currents (m s−1), significant wave height (m), and relative turbidity (normalized). Each sample is colored by month of collection, and shapes are based on reef location.

The canonical or CCA ordination was constrained by these four habitat variables, with CCA1 and CCA2 verified as significant by one-way ANOVA. Categorical data were analyzed vs. CCA1 and CCA2: month (R2 = 0.52, p = 0.001) and reef (R2 = 0.15, p = 0.009) were found to be significant, but the host species was not. Given that host species was not a significant factor to CCA, we analyzed samples from both species together. Accordingly, the CCA ordination plot annotated by month and reef (Fig. 6B) shows that microbial communities cluster primarily by month, with samples collected in March (N = 20) correlating with significant wave height and relative turbidity, and samples collected in September (N = 24) correlating with temperature; samples collected in June (N = 11) correlated with alongshore current speed.

Microbial taxa correlated with changes in waves and temperature

To identify individual microbial taxa that correlated with changes in the physical oceanographic habitat, we used a RFR for variable selection, and then applied a simple linear regression on individual variables. Based on ordistep model outputs, we analyzed four variables with RFR—significant wave height, relative turbidity, alongshore currents, and sea temperature. The RFR classification model was significant for significant wave height (p < 0.001, R2 = 0.85), alongshore currents (p < 0.01, R2 = 0.60), and sea temperature (p < 0.001, R2 = 0.85). The model was not significant for relative turbidity (p = 0.92, R2 = 0.001). The five microbial taxa with the highest importance values from each significant RFR model are listed in Table S1. That table also notes which of these “important” taxa were significantly correlated (p < 0.05, R2 > 0.2) with their respective habitat variables. For significant wave height, two taxa were significant from the order Flavobacteriales, and one from Marine Group II. For temperature, one taxon was significantly correlated from the order SAR86 (Fig. 7). There were no microbial taxa that were found to be correlated significantly with alongshore currents (i.e., none with p < 0.05 and R2 > 0.2).

Figure 7 Microbial taxa that correlated with physical habitat variables in Southeast Florida reefs.

Samples are colored by microbial order. The colored line represents the linear regression fit between the habitat variable (x-axis) and the microbe’s relative abundance (y-axis); gray shadow around the line is the 95% confidence interval. Plots show microbial taxa vs. physical habitat variables that were found to be significantly correlated to (p < 0.05, R2 > 0.2) (A) significant wave height (m) and (B) temperature (°C).

Discussion

In our study, we found distinct interactions of individual physical habitat variables with multiple microbiome structure variables. These physical oceanographic habitat variables do co-vary with one another over a range of time and space scales. For example, water temperatures can be driven by thermal advection, temperature gradients can force water flow through horizontal convection, and turbidity is often the result of wave breaking. Recent published results have demonstrated that indeed, the complex interaction between physical variables like flow, light, and temperature may potentiate their effects on coral microbiomes (Lee et al., 2017; Van Oppen et al., 2018). To add to the complexity of these physical habitat factors, a combination of physico-chemical components also works together to affect the microbial communities on coral reefs (Roik et al., 2016). However, here, we aimed to tease apart which univariate responses of the coral microbiome to individual habitat variables were significant.

Our results show that 95% of the variance in microbial beta-diversity was explained in the first two PCoA axis, and that coral species had the highest group separation (R2 = 0.38) among categorical factors (i.e., month, species, reef; Fig. 6A). Coral species (Porites astreoides and S. siderea), each have a unique microbial diversity composed of different taxa (Figs. 4 and 6A). However, we also found correlations across the two host species, between coral microbiomes and temperature, turbidity (related to available light), and multiple scales of waterflow (i.e., those related to modeled larger-scale ocean currents, smaller-scale responses to sea-level gradient, and waves).

We found that increases in tide height, sea-level, sea temperature, and modeled alongshore currents were associated with declines in microbial Shannon diversity and/or evenness (Fig. 5). Among shallow coastal environments such as those on tidal flats, tide changes affect the structure and function of the microbial community (Lv et al., 2016). Similarly, studies of reefs below the tidal zone have shown that differing seawater depths (related to available light and flow regimes) do correlate with changes in bacterial communities (Bonthond et al., 2018; Klaus et al., 2007; Pantos et al., 2015). Although we found increased sea-level and tide height both independently associated with decreased alpha-diversity, we cannot state which specific physical process explains these results. This is because the overall range of relative water height variations observed in this study from both modeled tides and direct (moored sea pressure) measurements was small, 0.25 m or less than 5% of the shallowest mean site depths (Fig. S2). Hence, any influence that sea-level (including tides) had on these microbiomes was more likely due to low-frequency variability in currents and/or transports forced by sea-level gradients, rather than to the sea-level variations themselves. In particular, low-frequency variability from both wind setup and tide can drive cross-shelf transport (Castelao et al., 2010; Lee & Smith, 2002; Leichter et al., 1996), affecting microbial connectivity (Säwström et al., 2016), flushing times (Castelao et al., 2010), and the availability of prey and nutrient inputs (Leichter, Stewart & Miller, 2003; Monismith et al., 2010; O’Connell et al., 2018). Such effects may serve to reduce microbial alpha-diversity on reefs (O’Connell et al., 2018) and may explain the correlations we observed.

For beta-diversity, the four habitat variables that best describe the data were significant wave height, alongshore currents, relative turbidity, and temperature. In March, microbial beta-diversity was correlated (Fig. 6B) with high multi-site median relative turbidity (Fig. 3C, all sites) and a range of significant wave heights between reefs (Fig. 3B, for example, Pillars vs. Oakland Ridge). Corals sampled in the present study were at bottom depths between 6 and 12 m and thus, were likely to be influenced by waves of the amplitudes we found from the model (Fig. 3; Fig. S3), both from orbital motions (Rogers et al., 2015) and wave-driven circulation (Rogers et al., 2018).

In response to significant wave height, we found that a euryarchaeotal taxon from Marine Group II decreased throughout the study period with increases in wave height (Fig. 7A). Euryarchaeota are known microbiome members in corals, with the majority of this taxon belonging to Marine Group II (Siboni et al., 2008). However, no study, to our knowledge, has described the role played by these archaea in coral microbiomes (Bourne, Morrow & Webster, 2016). Our analysis suggests that Marine Group II may play a larger role in the coral microbiome during times of lower wave activity such as June and, at some sites, in September. In March, during times of higher wave activity, there was an increase in relative abundance of the bacterial order Flavobacteriales, genus Tenacibaculum (Fig. 7A). The genus Tenacibaculum encompasses bacteria that are associated with the marine environment and can be pathogenic (Park et al., 2016; Smage et al., 2016). This Tenacibaculum ASV has a 100% sequence identity with Tenacibaculum sediminilitoris—an isolate from tidal flats (Park et al., 2016). Both tidal flat sediments and seawater are sources of Tenacibaculum; therefore, it is possible that the presence of this genus during high wave activity is due to sedimentation resuspension.

The CI dataset is designed to map out visible influences of inshore events on coral reefs using satellite ocean color data. This applies equally to plumes associated with the effects of breaking waves inshore, and those potentially attributable to human activities. High turbidity on the shelf can be due to waves through erosion or sediment resuspension. However, when turbidity occurs in the absence of higher waves (Figs. 3B–3C), as it did at some sites in March (Oakland Ridge and Emerald) and again in June (Barracuda and Oakland Ridge), it is considered more likely to be due to human activities such as dredging or flood control (Barnes et al., 2015; Gramer & Hendee, 2018). Indeed, a major port expansion was ongoing at Port of Miami, approximately nine km east-southeast of Emerald Reef, and 13, 37, and 46 km from Pillars, Barracuda, and Oakland Ridge, respectively. Primary dredging on this project ended on April 8, 2015, with “spot” dredging that continued into September of 2015. In March alone, we see historically high turbidity throughout the reefs we sampled (Fig. 3C), with a plume that includes the area surrounding the outer channel of Port of Miami (Fig. 2A). Interestingly, in March data, we also see a correlation between relative turbidity and coral microbiome beta diversity (Fig. 6B).

With regard to waves in September, we note that Tropical Storm Joaquin was meandering over waters near the Bahamas beginning on 28 September, and had intensified to a Category 4 hurricane by 1 October. During the final sampling day of the present study on 28 September, this storm appears to have already brought higher significant wave heights to the entire shelf, which affected samples taken at both Pillars and Emerald Reef (Fig. 3B). Modification of the reef environment by remote tropical systems occurs episodically in south Florida (Manzello et al., 2007). This result suggests that tropical weather systems impacted the physical environment of these reefs via wave action (Figs. 2 and 3). However, we note that waves from this indirect effect were still smaller than those from the direct effect of synoptic weather systems from the previous March (Fig. 3B, all sites). The effect in September was also only sampled at two of the four reefs, due to timing. Either of these differences could explain why microbiome structure related to waves stood out in March, but not in September (Fig. 6).

A number of studies have reported that the microbial community in corals responds to temperature fluctuations (McDevitt-Irwin et al., 2017; Zaneveld et al., 2016). In this study, we also see that coral microbial alpha- and beta-diversity correlated with temperature, with September samples in particular clustering closely together by temperature (Figs. 5 and 6B). In September, we see an increase in relative abundance in SAR86 from the class Gammaproteobacteria. This bacterium is found in oligotrophic environments and is known to respond to temperature changes (Dupont et al., 2012). It is unknown how this bacterium may affect the host’s health, but it may play an important role in Porites astreoides and S. siderea holobiont response during times of high temperature (Cárdenas et al., 2012; Pootakham et al., 2018).

The pattern of sea temperatures measured by the divers and in situ moorings in September appears consistent with summer-time upwelling superimposed on the annual “weather” cycle, rather than simple annual variability (Fig. 3A). Upwelling is known to occur episodically but frequently on the east Florida shelf (Pitts & Smith, 1997; Walker & Gilliam, 2013), and has been observed to affect temperature on reefs there at a wide range of depths (Gramer et al., 2018). Due to upwelling, some reefs in Florida can experience cooler water temperatures in August and September than they experience in January (Leichter, Helmuth & Fischer, 2006; Walker & Gilliam, 2013). In addition to modifying the thermal environment, upwelling can also enhance the flux of soluble nutrients onto a reef (Gramer et al., 2018). Furthermore, the onset and relaxation of an upwelling event may increase flushing of the shelf environment. A combination of these effects from upwelling may help explain the particular relationship between temperature and microbial community structure that we saw in the clustered data for September.

Conclusions

In conclusion, we identified physical oceanographic habitat variables that correlate with structure in the microbiomes of two hard coral species on four urban-impacted reefs. Our study, also identified summer upwelling at some of these reef sites, which decreased bottom sea temperatures and may explain some aspects of the observed microbiome structure. The study design, together with our use of regional-scale model and satellite data to characterize the physical habitat, did not allow for lower bounds to be determined on the significant (decorrelation) distances and time scales associated with variability in that habitat, nor the scales of the diversity in the coral microbiomes. Our study, nonetheless, indicates potentially important future considerations for improving coral health and outplant success in Florida reefs. This suggests that targeted, fine-scale future studies should be conducted at outplant sites, combining continuous physical measurements with repeated surveys and biological sampling, in order to further establish the relative importance of various oceanographic habitat characteristics in determining microbiology, coral health, and restoration success.

Supplemental Information

Supplemental Information 1 Linear regression of satellite relative turbidity (CI) correlated with in-water turbidity units (NTUs) taken by Acoustic Doppler current profiler (ADCP) (NTU data used in the Staley et al., 2017).

From from November 18, 2013—September 3, 2015, a cruise collected NTU data from Oakland Ridge reef, Barracuda reef, Pillars reef, and Emerald Reef. Only those in-water measurements meeting a sensitivity criterion (NTU > 0.5) were correlated to CI data to validate the model. NTU data was only used for validation of the model since the dates of tissue sampling did not correspond with the dates of ADCP data collections.

Click here for additional data file.

Supplemental Information 2 NOAA WaveWatch III mean output.

(A) For 2005–2015 for the northern portion of the sampling region. (B) Result of applying wave attenuation model using 10 m resolution bathymetry to the output of panel “A”.

Click here for additional data file.

Supplemental Information 3 Bathymetry-attenuated significant wave height [m] from NOAA WaveWatch III, time-averaged across the full weeks surrounding each sampling period.

(A)March, (B) June, and (C) September. The 12 sampling sites are denoted by different shapes as follows: squares represent Oakland Ridge reef, circles represent Barracuda reef, crosses represent Pillars reef, and triangles represent Emerald Reef.

Click here for additional data file.

Supplemental Information 4 Regressions between near-bottom sea temperature time series and FACE shallow mooring at seven m depth (x-axis), and minimum dive-watch temperature for each of the 55 samples in the present study (y-axis).

Reef location is indicated in the y-axis label of each regression plot, along with the distance from seven m mooring in km, and average sampling site depth for that reef.

Click here for additional data file.

Supplemental Information 5 The 14 day centered average of near-bottom sea pressure anomaly (“sea level”).

The FACE mooring at 7seven m depth (black line, time series; individual sample dates, colored stars).

Click here for additional data file.

Supplemental Information 6 Distributions (approximating probability density functions) for dynamical habitat variables considered in the present study.

A) Alongshore currents (1 d−1, m s−1), (B) cross-shore currents (1 d−1, m s−1), (C) near-bottom sea temperature at seven m mooring (3 h−1,°C), (D) near-bottom sea temperature at 26 m mooring (3 h−1,°C), (E) relative turbidity (approximately 1 d−1), (F) significant wave height (1 d−1, m). Each distribution includes all available data from the year 2015.

Click here for additional data file.

Supplemental Information 7 Top 5 microbial taxa selected by random forest regression analysis for three habitat variables.

Table lists linear regression pvalues and R2. Bold markings indicate ASV with p < 0.05, R2 > 0.2 that were plotted

Click here for additional data file.

Additional Information and Declarations

Competing Interests

Author Contributions

DNA Deposition

Data Availability

The authors declare that they have no competing interests.

Stephanie M. Rosales analyzed the data, prepared figures and/or tables, authored or reviewed drafts of the paper, approved the final draft.

Christopher Sinigalliano conceived and designed the experiments, performed the experiments, contributed reagents/materials/analysis tools, approved the final draft.

Maribeth Gidley conceived and designed the experiments, performed the experiments, contributed reagents/materials/analysis tools, approved the final draft.

Paul R. Jones performed the experiments, approved the final draft.

Lewis J. Gramer conceived and designed the experiments, analyzed the data, prepared figures and/or tables, authored or reviewed drafts of the paper, approved the final draft.

The following information was supplied regarding the deposition of DNA sequences:

Demultiplexed 16S rRNA sequences from the 55 samples are available at the National Center for Biotechnology Information (NCBI) Sequence Read Archive (SRA) database (#SRP076111).

The following information was supplied regarding data availability:

The biome data, ASV table, and metadata for this analysis are available at Figshare: Rosales, Stephanie; Sinigalliano, Christopher; Gidley, Maribeth L.; Jones, Paul; J. Gramer, Lewis (2019): Oceanographic habitat and the coral microbiomes of urban-impacted reefs. figshare. Dataset. DOI 10.6084/m9.figshare.7388672.

The code for the statistical analysis for the microbial data and the corresponding figures are available at figshare: Rosales, Stephanie (2019): Jupyter notebook for microbiome analysis of “Oceanographic habitat and the coral microbiomes”. figshare. Software. DOI 10.6084/m9.figshare.7925495.

The code used to generate physical ocean habitat variables and the corresponding figures are available at Figshare J. Gramer, Lewis (2019): Physical habitat code. figshare. Software. DOI 10.6084/m9.figshare.7925546.

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
