# Peer review of "Oceanographic habitat and the coral microbiomes of urban-impacted reefs"

_PeerJ, doi:10.7717/peerj.7552_

## Round 0.1 · original submission · Major Revisions

I have now obtained two thorough reviews of your paper and am happy to report that both reviewers found your paper of high interest and well written. Both, however, had a number of substantive comments for you to consider towards improving your paper. My feeling is that the paper lies somewhere between a Major and Minor revision. Note Reviewer 1 is suggesting some additional analyses and a bit deeper dive into the data. Thus, I have recommended 'major' revision, but I think everything is very doable.

·

Basic reporting

The article is generally well written with clean, professional figures.

The data used are publicly available from the NCBI Short Read Archive, but it would be nice to mention to readers where to find the accession numbers for the samples or runs as well (i.e. SRR3632944, SRR3632945, etc) if they were looking to reproduce the analyses.

There also seems to be extra data (32 extra samples) other than that used in the study provided, at least in the provided ASV feature table where I looked. I'm not sure why this is present, perhaps the authors could clarify or perhaps I missed something. The ASV feature table also reports the minimum feature count as 434 rather than the 403 reported (Line 154), and the number of ASVs as 26918 rather than 27692 reported (Line 334). I'm not sure where these discrepancies come from.

I do find the combined presentation of the Shannon diversity metric with taxonomic composition in Figure 4 to be a bit unusual and confusing at first glance. I would prefer that the taxonomic composition be presented separately such that all the bars are scaled to 1.0 or 100% to so that readers can compare the relative abundance of individual taxa across sites and months more easily.

Figure 5 is busy—I find the interpretation of % coral cover (mapped to symbol size) especially difficult because of the different symbols used.

Figure 6a would be better if it showed the month of sampling and/or identity of the site for each sample.

Figure 7 has overlapping points, reducing the opacity of points would improve the ability to discern separate points.

Experimental design

The article is well within the scope and aims of PeerJ.
The authors present a MiSeq 300-bp dataset consisting of 55 samples. Assuming that a standard Illumina v2 300-bp reagent kit, approximately 15 million reads or read pairs are produced. The data were rarefied down to 403 counts per sample for a total of (403 * 55) = 22,163 sequence variants. Thus, roughly 99% of the data from the Illumina run was not utilized. Additionally, 403 reads seems like a small number—it's possible that study could be better off with the smaller libraries thrown out.

There are better alternatives to throwing away nearly all of the data through rarefaction (see McMurdie & Holmes, 2014). The authors may also find Amy Willis' discussion on rarefaction and alpha diversity in microbial ecology useful: https://www.biorxiv.org/content/10.1101/231878v1 . The small library size of the smallest samples and the rarefaction likely mean that alpha diversity analysis is prone to bias, or at the very least an underestimation of the true alpha diversity.

The authors might also be interested in Gloor et al., 2017, which suggests additional methods for dealing with compositional microbiome data, but implementing the methods suggested by Gloor et al., 2017 (e.g. CLR, ILR, ALR, Aitchison) is not necessary for the acceptance of manuscripts in journals at this time.

The general reproducibility of the paper could be improved with the inclusion of the QIIME2 & DADA2 scripts. Further information on this part of the analysis would also be appreciated to check. I am interested in checking that the primer regions of the reads were appropriately fully removed before processing with DADA2. I am also interested to know the rationale for the trimming that was done in general, since it doesn't make sense to me right off the bat. DADA2 is able to handle low quality bases, so trimming off of the ends of reads isn't strictly required as in other amplicon workflows, but does provide a boost to the speed of processing the reads. In my opinion, the reproducibility of this work & usefulness for future work would also be improved by providing the sequence variants found—this is one of the greatest benefits to an ASV-based pipeline: the exact variants can be reported and compared across studies. Including two files, the taxonomy file containing the taxonomic lineage information for each sequence variant and a FASTA file containing the sequence of each ASV would likely make this study more amenable for future use by others. Including the Newick tree file also would make the code in the Jupyter notebook reproducible, but currently it is not easily reproducible.

I would like to see how the authors arrived at the statement of > 95% of the variance being explained by host species in Line 418. If this is from adding up the first two axes of their PCoA, I don't think this interpretation of PCoA axes (more specifically the complete attribution of the % variance of each PCoA axis to host species) is correct. A PERMANOVA might be more appropriate for determining how much of the variance is explained by host species.

Gloor GB, Macklaim JM, Pawlowsky-Glahn V, Egozcue JJ. 2017. Microbiome datasets are compositional: And this is not optional. Frontiers in Microbiology 8:1–6. DOI: 10.3389/fmicb.2017.02224.
McMurdie PJ, Holmes S. 2014. Waste Not, Want Not: Why Rarefying Microbiome Data Is Inadmissible. PLoS Computational Biology 10. DOI: 10.1371/journal.pcbi.1003531.

Validity of the findings

It may be that the findings still hold after re-analysis of the data or the authors may even find additional patterns, but I would like to see the dataset re-analyzed to address the comments I made in the "Experimental Design" section, primarily with the use of a method of accounting for library size differences without the use of rarefaction.

Additional comments

Overall, the manuscript appears to be in pretty good shape, and I would most likely expect it to be acceptable after the suggested changes.

Reviewer 2 ·

Basic reporting

Authors used clear English throughout the manuscript.
Authors clearly stated their objectives and hypothesis.
The manuscript is clearly written, with proper citations, and provides sufficient background.
It is especially noted, and appreciated, that authors provide helpful context, references, and explanations for physical data/phenomena that may not be familiar to biologists or non-experts.
The article is well-structured, reads logically, and flows well.
Tables and figures are helpful and formatted well.
Raw data is archived in publicly available databases.

Experimental design

The research is original – studies regarding the impacts of physical parameters on microbiomes is limited and the data presented here help to fill that knowledge gap.
Researchers appear to have included sufficient replication in their experimental design by collecting replicate tissue samples from multiple sites (n=3) within multiple reefs (n=4). However, the number of colonies sampled per site should be clearly stated in the methods section.
I understand these samples and data were collected as part of a previously published study; however, it is unclear why more fine scale depth data were not collected by divers. Also, it seems reef water samples, and specifically turbidity data for reef water, were collected in the previously published study by Staley et al. 2017. Is it possible to ground truth the satellite derived turbidity data with these data, or if I am not mistaken in the details of the collection, use the reef water turbidity data in the current analysis?

Validity of the findings

Thorough statistical analysis was employed with the data. However, being that the data is mainly observational and not experimental data, it is difficult to control for multiple environmental variables. That being said, the authors did a good job in honestly assessing the limitations of their data and discussing their findings within the appropriate context.
One topic that was absent in the discussion but that should be further explored is the likelihood that a combination of physical and chemical properties are shaping microbiomes. Again, it seems some seawater chemical nutrient data were collected for the original study and, if appropriate/relevant, this manuscript might benefit from including those data.
Conclusions were clearly linked back to their original research question.
Authors did not over-speculate when interpreting and discussing their findings.

Additional comments

Overall, this was a very well-written manuscript. It read logically and flowed easily. It is clear that the authors paid close attention to detail in all aspects of the manuscript. There were also very few spelling/grammatical/etc. errors. The following are suggested edits/comments for the authors:

Introduction:
Line 74: Do the authors mean to reference opportunistic bacterial pathogens here?
Line 75: Replace “effected” with “affected”
Line 79: Please provide a citation

Methods:
Lines 122-123: Please detail how diver-collected temperature data were recorded. I did not locate this information in the publication or supplemental materials in Staley et al. 2017 (Later I see “dive-watch sea temperature” is mentioned in lines 258-259. Please also include this in the methods section)
Lines 125-128: Please list how many colonies were sampled from each site (or at least refer the reader to Table 1 for that info). Also, detail whether or not two polyps per coral head were combined to equal one sample or analyzed separately (based on the numbers in Table 1 I’m assuming they were combined, but this should be clear to the reader and explained in the methods).
Line 133: How was the skeleton sampled with a syringe? In Staley et al. 2017 the methods detailed in the supplemental information state “Coral tissue was collected by syringe biopsy, where the syringe was held firmly against the coral skeleton and the plunger pulled back, causing a suction that draws the entire coral polyp into the syringe (while causing no damage to the coral skeleton)…” Also, Were polyps macerated/homogenized prior to filtration?
Lines 143-148: Further detail is needed regarding the bioinformatic analysis. For example, how many sequences were obtained per sample? What was the minimum merge length? What type of quality filtering was used? Average sequence length? What type of clustering was used to determine ASVs? How were chimeras and singletons addressed?
Lines 188-191: Unclear as to why specific depth data wasn’t collected by divers when individual colonies were sampled. Please explain the rationale for deriving seafloor depths from NOAA’s data.

Results:
Lines 254-258: Recommend moving this information to the methods sections
Figure 6B: Two of the vector labels are overlapping and difficult to read.

---

## Round 0.2 · accepted · Accept

Thank you for your careful revisions in response to the previous reviews. I sent your paper back to both reviewers and they are both satisfied with your responses and adjustments. They both really like the study and think it is well done. They both provided a few minor things you'll want to take a look at as you prepare your final version while in production.

·

Basic reporting

I greatly appreciate the effort that authors made in implementing the suggestions made.

If the authors used ggplot to produce figure 6A, the 'ggrepel' library could be used to quickly improve the readability of the text labels.

Minor typos:
Line 127: "scrapping" should presumably be scraping?
Line 171: "asses"

Experimental design

I also greatly appreciate the authors addressing my concerns about rarefaction in their study by clarifying the language used and testing the sensitivity of their data to rarefaction by running the analysis with the samples with the lowest numbers of reads dropped.

While I still hold concerns that, in general, rarefaction is not ideal for alpha-diversity estimation, in light of the the robustness of the findings to rarefaction at a higher number, I do believe that the patterns described are true and find the evidence sufficient for publication.

Validity of the findings

The findings were robust to re-analysis of the data without the two samples w/ very low read counts.

Additional comments

WIth the changes made, I find this manuscript suitable for acceptance. It is evident that a lot of work went into this study.

I did make some minor suggestions in Section 1: Basic Reporting, which could be easily fixed before the final version, but I don't find that these changes would merit a recommendation of 'minor revisions,' so I am recommending that the article be accepted at this stage. The authors may choose to make those changes if they see fit.

Reviewer 2 ·

Basic reporting

Authors used clear English throughout the manuscript.
Authors clearly stated their objectives and hypothesis.
The manuscript is clearly written, with proper citations, and provides sufficient background.
It is especially noted, and appreciated, that authors provide helpful context, references, and explanations for physical data/phenomena that may not be familiar to biologists or non-experts.
The article is well-structured, reads logically, and flows well.
Tables and figures are helpful and formatted well.
Raw data is archived in publicly available databases

NOTE: Upon reviewing the figshare files, I did not see that the QIIME2 code was uploaded as indicated. Please upload this information as soon as possible.

Experimental design

The research is original – studies regarding the impacts of physical parameters on microbiomes is limited and the data presented here help to fill that knowledge gap.
Researchers appear to have included sufficient replication in their experimental design by collecting replicate tissue samples from multiple sites (n=3) within multiple reefs (n=4). However, the number of colonies sampled per site should be clearly stated in the methods section.
SATISFACTORILY ADDRESSED BY AUTHORS

I understand these samples and data were collected as part of a previously published study; however, it is unclear why more fine scale depth data were not collected by divers. Also, it seems reef water samples, and specifically turbidity data for reef water, were collected in the previously published study by Staley et al. 2017. Is it possible to ground truth the satellite derived turbidity data with these data, or if I am not mistaken in the details of the collection, use the reef water turbidity data in the current analysis?
SATISFACTORILY ADDRESSED BY AUTHORS

Validity of the findings

Thorough statistical analysis was employed with the data. However, being that the data is mainly observational and not experimental data, it is difficult to control for multiple environmental variables. That being said, the authors did a good job in honestly assessing the limitations of their data and discussing their findings within the appropriate context.
One topic that was absent in the discussion but that should be further explored is the likelihood that a combination of physical and chemical properties are shaping microbiomes. Again, it seems some seawater chemical nutrient data were collected for the original study and, if appropriate/relevant, this manuscript might benefit from including those data.
AUTHORS ADDRESS THIS CONCERN BY PROVIDING A REFERENCE ON THE ROLE OF PHYSIO-CHEMICAL PROPERTIES ON SHAPING MICROBIOMES, BUT I THINK A SENTENCE OR TWO TO EXPAND ON THIS, OR PROVIDE AN EXAMPLE, IS NEEDED.

Additional comments

ALL OTHER COMMENTS SATISFACTORILY ADDRESSED BY AUTHORS